# Profile of Bacterial Community and Antibiotic Resistance Genes in Typical Vegetable Greenhouse Soil

**DOI:** 10.3390/ijerph19137742

**Published:** 2022-06-24

**Authors:** Xuexia Yuan, Yong Zhang, Chenxi Sun, Wenbo Wang, Yuanjuan Wu, Lixia Fan, Bing Liu

**Affiliations:** 1Institute of Agricultural Standards and Testing Technology for Agri-Products, Shandong Academy of Agricultural Sciences & Shandong Provincial Key Laboratory of Test Technology on Food Quality and Safety, Jinan 250100, China; chelseasnn@163.com (C.S.); wangwb74@163.com (W.W.); wuyj1977@163.com (Y.W.); superdemeter@163.com (L.F.); 2Shandong Provincial Land Surveying and Planning Institute, Jinan 250014, China; zhangyongjeff@163.com; 3Resources and Environment Innovation Institute, School of Municipal and Environmental Engineering, Shandong Jianzhu University, Jinan 250101, China

**Keywords:** antibiotic resistance genes, bacterial community, *IntI* gene, heavy metal, vegetable greenhouse

## Abstract

The use of vegetable greenhouse production systems has increased rapidly because of the increasing demand for food materials. The vegetable greenhouse production industry is confronted with serious environmental problems, due to their high agrochemical inputs and intensive utilization. Besides this, antibiotic-resistant bacteria, carrying antibiotic-resistance genes (ARGs), may enter into a vegetable greenhouse with the application of animal manure. Bacterial communities and ARGs were investigated in two typical vegetable-greenhouse-using counties with long histories of vegetable cultivation. The results showed that *Proteobacteria*, *Firmicutes*, *Acidobacteria*, *Chloroflexi*, and *Gemmatimonadetes* were the dominant phyla, while *aadA*, *tetL*, *sul1*, and *sul2* were the most common ARGs in greenhouse vegetable soil. Heatmap and principal coordinate analysis (PCoA) demonstrated that the differences between two counties were more significant than those among soils with different cultivation histories in the same county, suggesting that more effects on bacterial communities and ARGs were caused by soil type and manure type than by the accumulation of cultivation years. The positive correlation between the abundance of the *intI* gene with specific ARGs highlights the horizontal transfer potential of these ARGs. A total of 11 phyla were identified as the potential hosts of specific ARGs. Based on redundancy analysis (RDA), Ni and pH were the most potent factors determining the bacterial communities, and Cr was the top factor affecting the relative abundance of the ARGs. These results might be helpful in drawing more attention to the risk of manure recycling in the vegetable greenhouse, and further developing a strategy for practical manure application and sustainable production of vegetable greenhouses.

## 1. Introduction

Soil microorganisms, one of the key parts of the soil ecosystem, play important roles in nutrient cycling and fertility maintenance, such as by regulating levels of C, N, and P [1]. Soil bacterial communities contribute to the health of flora and fauna directly or indirectly in terrestrial ecosystems. The abundance and diversity of soil bacterial communities are widely used as effective indicators of soil health because of their high sensitivity to environmental factors and soil nutrient status [2,3]. Antibiotic-resistant bacteria and their antibiotic-resistance genes (ARGs) are regarded as one of the most significant emerging contaminants, due to their threats to environmental ecosystems and public health [4,5]. Multiple antibiotic-resistant bacteria and ARGs have been reported in various environments, including lake [6], river [7,8], pasture [9], agriculture [10], and air [11] environments. Soil is a major reservoir and sink of both antibiotics and ARGs. Except for intrinsic resistance, one of the sources of ARGs in soil has been a large amount of antibiotic input through manure fertilization, which is a common method of promoting soil fertility and nutrients for plant growth. Antibiotics have been reported as a driver of ARG enrichment, because the selection pressure on soil bacteria by metabolic changes and the enzyme activity of microorganisms produce ARGs [12]. Besides this, ARGs can be disseminated into plant root and leaf endophytes from manure-fertilized soil [13,14]. Soil–microbe–animal–plant systems became a vital nexus that could influence human health via direct/indirect connection. Therefore, ARGs in soil, particularly in manure-fertilized soil compartments, have become an increasing hotspot globally.

In addition, environmental factors (heavy metals and physicochemical properties) contribute to bacterial communities and ARGs. Heavy metal, even in low concentrations, can accelerate the evolution and spread of ARGs in environments through coresistance and/or cross-resistance by rebuilding the structure of microbial community [15,16]. For example, the majority of bacteria are significantly positively correlated with As, Cd, Cr, Ni, and Pb, while being negatively correlated with Cu, Mn, and Zn in chicken-manure-fertilized soil [17]. Cu, Zn, and Pb are significantly positively correlated with ARGs after 30 years of swine manure exposure in agricultural soil [18]. As and Cd were found to be the predominant factors shaping the distribution patterns of ARGs in paddy soils, by affecting the bacterial abundance [19]. Organic matter (OM) is one of the top soil properties affecting ARGs, and is negatively correlated with the abundance of ARGs. The soil pH and Zn content are the biggest contributors to the distribution pattern of ARGs in cinnamon soils [20].

The Chinese greenhouse vegetable production system has been boosted more than eight-fold from 1990 to 2014, making China the world’s main greenhouse vegetable producer [21]. Environment factors inside the greenhouse, including irrigation, temperature, humanity, etc., are significantly different from those in open fields. Additionally, increasing evidence has confirmed the presence of soil pollution and deterioration of soil quality, caused by frequent organic and chemical fertilizer, pesticide, and fungicide applications in greenhouses [22]. The concentration of antibiotics in greenhouse soils is also higher than that in open-field soils [23]. All the above factors probably would affect the profile of bacterial communities and ARGs in soil. Therefore, more attention should be paid to bacterial communities and ARGs in greenhouses and their potential risks to food safety.

An increasing number of studies have focused on bacterial communities and ARGs in vegetable greenhouses. One study predicted that soil microbial communities are influenced by heavy metal pollution in greenhouses, which was investigated through 16s rRNA sequences analysis [22]. Abundances of *tetW*, *tetO*, *sul1*, and *sul2* were significantly increased in manure-amended greenhouse soil [24]. Another study found no significant difference of ARGs in soil amended with different types of manure, whereas the diversity and abundance of ARGs were elevated in fertilized soil [25]. The bacterial communities and ARGs, and their relationships with environmental factors, are still relatively limited, particularly in greenhouses with long histories of over 15 years. Shandong is a main greenhouse cultivation province, and counts for half of the vegetable cultivation area in China [24]. Shouguang, in Shandong province, has the largest vegetable production industry and wholesale market in China. Anqiu, another city with high vegetable production, is famous for its export of vegetables. Both of them have long histories of manure application into greenhouses. Here, soils from two typical greenhouse production counties with long-term manure-application histories were selected to investigate (1) the profile of bacterial communities, ARGs, and integron–integrase (*intI*) genes in vegetable greenhouse soils; (2) the relationships of bacterial communities, ARGs, and *intI* genes with physicochemical properties and heavy metals; (3) the potential bacterial hosts harboring ARGs and *intI* genes. The results will be helpful in gaining a deeper understanding of the bacterial communities and ARGs in long-term greenhouse cultivation with manure applications, and in providing scientific support to make a strategy for practical manure application and sustainable production of vegetable greenhouses.

## 2. Materials and Methods

### 2.1. Sampling Sites and Soil Sampling

Sampling sites were located in two counties, Shouguang (SG) (118°76′ N, 36°77′ E) and Anqiu (AQ) (119°32′ N, 36°22′ E), Shandong province. Samples of six vegetable greenhouse soils, classified as cinnamon soil, were collected in SG within a range of 600 m. These sites have had continuous vegetable greenhouse cultivation with manure application for 3 years (SG1), 5 years (SG2), 7 years (SG3), 10 years (SG4), 15 years (SG5), and 17 years (SG6). The vegetable planted was eggplant. Poultry manure was used as a base fertilizer annually. The cropping period was from August to June the next year. Samples of five vegetable greenhouse soils classified as fluvo-aquic soil were collected in AQ within a range of 300 m. These sites have had continuous vegetable greenhouse cultivation with manure application for 3 years (AQ1), 7 years (AQ2), 10 years (AQ3), 14 years (AQ4), and 16 years (AQ5). The crops planted were cucumber and tomato. Swine manure and cattle manure were used as base fertilizers annually. The cropping period was from October to the following June.

Soil samples were taken at the beginning of June. The area of each greenhouse was about 1300 m^2^. In each greenhouse, three plots were separated as replicated. Within each plot, ten subsamples were collected from the topsoil (around 0–20 cm) and then mixed thoroughly to obtain uniform representativeness. The soil was placed in sterile plastic bags and immediately transported in a cooling box with ice. In the laboratory, a portion of soil was stored at −80 °C for molecular analysis for ARGs, *intI* genes, and bacterial communities. A portion of soil was air-dried and sieved to smaller than 200 μm particles for physicochemical property and heavy metal determination. A portion of soil was freeze-dried under vacuum (48~72 h) and sieved through a 0.42 mm mesh for antibiotic determination.

### 2.2. DNA Extraction

FastDNA^®^SPIN Kit for Soil (MP Biomedicals, Santa Ana, CA, USA) was used to extract the total DNA from the soil [18]. A NanoDrop 2000 spectrophotometer (Thermo Fisher Scientific, Waltham, MA, USA) was used to examine the concentration and quality of the DNA. The qualified DNA was stored at −80 °C until analysis.

### 2.3. Illumina High-Throughput Sequencing

The soil bacterial 16S rRNA gene (V3-V4 hypervariable region) was amplified using the primer sets of 338F/806R (5′-ACTCCTACGGGAGGCAGCA-3′; 5′-GGACTACHVGGGTWTCTAAT-3′). The PCR reactions and conditions were performed as described by the previous study [26]. Purified and pooled PCR products were then pyrosequenced using the Illumina Hiseq 2500 platform (Biomarker Technologies Co., Ltd., Beijing, China). Raw Illumina fastq files were merged by Flash software (version 1.2.11) to obtain raw tags, and then were filtered by Trimmomatic software (version 0.33). Chimeric sequences were removed by UCHIME (version 8.1), and high-quality tags were obtained. The remaining sequences were then clustered into operational taxonomic units (OTUs) with similarity of ≥97% by employing USEARCH (version 10.0). OTU taxonomy classification was performed for each phylotype by using the RDP classifier based on the SILVA database (http://www.arb-silva.de (accessed on 27 September 2020) [27].

### 2.4. qPCR of ARGs and intI Genes

We employed qPCRs to determine the abundances of 18 selected genes. The information and details of primers, annealing temperature, and amplification size were listed in Appendix A. Each qPCR was conducted on a Lightcycler 480 II (Roche, Basel, Switzerland). The 20 μL qPCR reaction mixture consisted of 10 μL of SYBR Premix Ex Taq^TM^ Kit (TaKaRa, Shiga, Japan), 0.2 μL of each primer (20 µM), 2.0 μL of template DNA, and 7.6 μL of RNA-free water. Melting curve analysis was added at the end of each qPCR run to confirm the specificity of the amplification. We used 10-fold serial dilutions of extracted plasmids carrying the selected genes as standards for qPCR quantification according to the details of the previous study [18]. The copies of 16S-rRNA were determined to calculate the relative abundance of ARGs, which could be used to normalize the ARGs and *intI* genes abundance to the total bacterial DNA [7].

### 2.5. Determination of Physicochemical Properties and Heavy Metal Concentration

Soil pH was determined by pH meter after suspension of soil-to-water (W_soil_/V_deionized water_ = 1:2.5). Soil organic matter (OM) was determined after oxidation of potassium dichromate–sulfuric acid solution. A total of eight typical heavy metals were digested using HNO_3_-HCl-HClO_4_. The concentrations of Cd, Cr, Cu, Ni, Pb, and Zn were measured by inductively coupled plasma optical emission spectrometry (ICP-OES) and inductively coupled plasma mass spectrometry (ICP-MS). The concentration of As and Hg were measured by atomic fluorescence spectrometry (AFS) [28].

### 2.6. Determination of Antibiotics

A total of 17 antibiotics were analyzed in this study, including amoxicillin, ampicillin, cefoxitin, ceftiofur, chloramphenicol, florfenicol, flupenicolamine, gentamicin, streptomycin, sulfisoxazole, sulfamethoxazole, sulfadimidine, tetracycline, oxytetracycline, chlorotetracycline, doxycycline, and trimethoprim. The determinations of the 17 antibiotics were conducted as described in the study [29]. Briefly, the antibiotics in the soil were extracted using a buffer of methanol: acetonitrile: citrate (1:1:2). The extracted solutions were then purified and concentrated using Oasis HLB cartridges. After nitrogen-drying and concentrating, the final extracts were measured by an ultra-high-performance liquid chromatography-tandem mass spectrometry (UHPLC-MS/MS) system.

### 2.7. Statistical Analysis

The microbial α diversity index was calculated using Mothur (version 1.30). Principal coordinate analysis (PCoA) and variation partitioning analysis (VPA) were performed using the R package ‘vegan’. The redundancy analysis (RDA) was performed using Canoco5. Network analysis was drawn using Gephi (version 0.9.2) based on Spearman’s correlation analysis (*p* < 0.05).

## 3. Results

### 3.1. Characterization of Bacterial Community

As shown in Figure 1, the result of high-throughput sequencing of soil bacterial 16S rRNA genes revealed that the dominant bacterial phylum in AQ was *Proteobacteria* (around 0.311–0.376), followed by *Firmicutes* (around 0.087–0.287), *Acidobacteria* (around 0.113–0.191), *Chloroflexi* (around 0.045–0.089), *Gemmatimonadetes* (around 0.054–0.078) and *Actinobacteria* (around 0.057–0.097). In SG, the dominant bacterial phyla were *Proteobacteria* (around 0.236–0.295) and *Firmicutes* (around 0.154–0.299), then followed by *Acidobacteria* (around 0.122–0.198), *Chloroflexi* (around 0.094–0.166), *Gemmatimonadetes* (around 0.062–0.149). Generally, the abundances of *Proteobacteria* and *Actinobacteria* in AQ were higher than those in SG. On the contrary, the abundance of *Chloroflexi* in AQ was lower than that in SG. A heatmap of the bacterial communities showed the detailed differences at phylum level (Figure 2A). PCoA of the bacterial communities showed that there were more differences between two counties than between sites in the same county (Figure 2B).

A α diversity index including ACE, Chao1, Simpson, and Shannon demonstrated irregular changes both in AQ and SG (Appendix A). For ACE, Chao1, and Simpson, no significant difference was found between AQ and SG. Shannon in AQ was significantly higher than that in SG (*p* = 0.012), which indicated that bacterial community diversity in AQ was higher than that in SG.

### 3.2. Abundances of ARGs and intI Gene

Relative abundances of 14 ARGs detected in different greenhouse soils are shown in Figure 3. The relative abundances of *aadA* were the highest among all the 14 ARGs, and the values ranged from around 0.033–0.073. The next-most-abundant ARGs were *tetL*, *sul1*, and *sul2.* The four ARGs abovementioned were identified as the dominant genes, which accounted for around 87.91–94.34% of the total ARGs. The remaining 10 ARGs detected were in relatively much lower quantities. In AQ, the highest total relative abundance was found in AQ3, while the lowest was in AQ2. In SG, the highest total relative abundance was found in SG1, while the lowest was in SG4. These indicated that abundance of ARGs was not increased with the cultivation time increase. Generally, total relative abundance of ARGs in SG soils was higher than that in AQ soils, except for in AQ3. Heatmap results showed the difference among the treatments (Figure 4A). Furthermore, PCoA analysis showed that there were more differences between soils of two counties than among soils of different cultivation histories in the same county (Figure 4B). These indicated that ARGs were affected by soil type and manure type more than by accumulation of manure application time.

Two *intI* genes, *intI1* and *intI2*, were also investigated to measure the horizontal transfer potential of ARGs. Appendix A showed that the relative abundance of *intI1* was significantly higher than that of *intI2*. The total abundance of the *intI* gene ranged from 6.79^−4^ to 7.80^−3^. The highest abundance of *intI* in AQ was found in AQ3, and the lowest was in AQ5. In SG, the highest abundance of *intI* was found in SG2, and the lowest was in SG1. These also indicated that the *intI* genes were not accumulated as the year of vegetable cultivation passed.

Network analysis showed that correlation among ARGs was detected. For example, *cmlA* was found to be positively correlated with *aadA2*, and both of them were positively correlated with *sul1*, *sul2*, and *tetG.* Additionally, the abundance of *intI1* was positively correlated with *cmlA*, *aadA2*, *sul1*, *sul2*, *tetC*, *tetG*, and *tetL*, and the abundance of *intI2* was positively correlated with *strA*, *strB* (Figure 5A), which confirmed the horizontal transfer potential of these ARGs.

According to Figure 5B, 11 phyla were revealed that were totally positively correlated with ARGs, indicating that these phyla might be the potential hosts of specific ARGs. For example, positive correlations of *Gemmatimonadetes* with *cmlA*, *sul1*, *and sul2*, *Chloroflexi* with *sul1* and *tetM*, *Proteobacteria* with *tetB(P)*, and *Firmicutes* with *tetA* were found respectively. *Actinobacteria*, *Deinococcus-Thermus*, *Dependentiae*, *Hydrogenedentes*, *Tenericutes*, *FBP*, and *Spirochaetes* were also positively correlated with individual ARGs, though their abundances were relatively low. More attention should be paid to the positive correlation of *intI1* and *Gemmatimonadetes*.

### 3.3. Factors Influencing ARGs and Bacterial Community

Physicochemical properties and heavy metal concentrations in different vegetable greenhouse soils are shown in Appendix A. Only doxycycline was detected in soil from SG1, with the value of 12.69 µg∙kg^−1^. Therefore, physicochemical properties and heavy metal concentrations were used to further RDA analysis. RDA results are shown in Figure 6, which were used to assess the relationships among environmental parameters and bacterial communities (at phylum level). RDA1 and RDA2 could explain 33.45% and 23.11% of bacterial communities respectively. As demonstrated by the lengths of the variables in RDA, Ni and pH were the strongest environmental factors determining the overall bacterial communities. The contribution of effect of Ni on bacterial communities was 30.6%, which was followed by pH, Cd, OM, Cu, Zn, and As with the contribution of 17.7%, 13.5%, 12.1%, 7.9%, 6.5%, and 4.9%, respectively (Table 1). Ni, as well as Cr and Pb, were positively correlated with the dominant phylum *Proteobacteria*, whereas they were negatively correlated with the second dominant phylum *Fimicutes* and the fourth dominant phylum *Chloroflexi*. Soil pH was found have a positive correlation with the third dominant phylum, *Acidobacteria*.

As shown in Figure 7 and Table 1, RDA1 and RDA2 could in total explain 65.07% of the ARG profile, with 39.02% and 26.05% respectively, which meant that physicochemical properties and heavy metal concentrations could mostly determine the existence of ARGs. Among the factors, Cr was the top factor affecting the relative abundance of ARGs with a contribution of 29.2%, which was followed by Cu, Cd, OM, pH, Ni, and Zn with contributions of 12.2%, 12.4%, 11.7%, 10.3%, 8.0%, and 7.6%, respectively. Cr, as well as Ni, Pb, Cu, As, and organic matter, were positively correlated with *tetB(P)*, while they were negatively correlated with most of the ARGs detected, including *cmlA*, *aadA*, *aadA2*, *sul1*, *sul2*, *tetA*, *tetL*, and *tetM.* The effect of pH was just the opposite.

## 4. Discussion

### 4.1. Bacterial Community and Correlation with ARGs

*Proteobacteria*, *Firmicutes*, *Acidobacteria*, *Chloroflexi*, and *Gemmatimonadetes* were the main phyla found, which was similar to other reports on vegetable greenhouse soils [30]. However, the sequence of the relative abundance of these phyla in soil from different places was a little different. *Proteobacteria* might be simulated by the input of large amounts of organic and chemical fertilizer in greenhouses, because it was favored by conditions with high nutrient and carbon content [31]. There were more differences in bacterial communities between two counties than between different numbers of vegetable cultivation years. It is noted that the tendencies of effects on bacterial communities were similar to the results of the ARGs. This might be due to the factors affecting ARGs via changing various ecological niches for different bacterial populations, which result in the shift of microbial communities [32]. There was a study that showed that long-term greenhouse cultivation had no effects on microbial diversity [33], while another study found that microbial communities were significantly affected by greenhouse cultivation years [34]. Our study found that the abundance and diversities were significantly different, with no regular changes along with the increase of vegetable cultivation years.

### 4.2. Abundances of ARGs and intI Gene

Aminoglycoside ARGs were reported to be the most important ARGs in agricultural soil amended with animal manure [20,35]. The *aadA* gene encoding aminoglycoside nucleoside transferase was found to be the predominant ARG among all the ARGs in this study. Additionally, *aadA2*, another aminoglycoside nucleotide transferase gene, and two aminoglycoside phosphotransferase genes (*strA* and *strB*) found in this study further confirmed the prevalence of aminoglycoside ARGs. The widely usage of aminoglycoside antibiotics in animal husbandry since the late 1950s [7] might be responsible for the high level of microbial resistance [35] and harboring of ARGs against aminoglycoside [20]. The integrons having *aadA* or variant of *aadA* were located on either chromosomes or plasmids [36], which increases the future risk by wide dissemination. The gene *tetL*, encoding an efflux pump against tetracycline antibiotic, was much more abundant than other *tet* genes, which is consistent with previous reports in agricultural soils with long-term manure application [37]. Considering that a considerably higher abundance of *tetL* gene was detected in organic manure [38], and *tetL*-carrying plasmids (pBSDMV46A, pSU1, and pDMV2) persisted [39], the high prevalence and robust adaptability of *tetL* in manure-amended soil was reasonable. Two plasmid-borne sulfonamide ARGs (*sul1*, *sul2*), encoding the dihydropteroate synthase enzymes, were also dominant genes in vegetable greenhouse soil. The genes *sul1* and *sul2* were observed to occur together frequently [40], because *sul2* is essentially one of the variants of the *sul1* gene. However, *sul1* is often located in integrin gene cassettes, while *sul2* is commonly located in small *IncQ* family plasmid, which means that different abundances exist in different soil environments [41].

There was no regular change or continuous proliferation of ARGs along with repeating annual application of animal manure. Previous studies found similar results, showing that the abundance of ARGs increased irregularly with the years of manure application increase [20,24]. However, the elevation abundance of ARGs by manure exposure disappeared gradually within a crop-growing season due to the resilience capacity of soil [42]. Actually, the ARGs profile was a comprehensive result of several behaviors, including prevention by indigenous microorganisms of the invasion from fecal microorganisms [43], horizontal gene dissemination from fecal bacteria to indigenous bacteria [36], and stability of extracellular ARGs [44]. The levels of diversity and relative abundance of ARGs in the cinnamon soils were lower than those in the saline–alkali and fluvo-aquic soils in vegetable greenhouse [20]. Cattle manure and swine manure contained different diversities and abundances of ARGs [18]. Significantly different abundances of ARGs among cattle manure, swine manure, and poultry manure, and further ARGs in the soils amended with different types of animal manure, were observed [45]. Thus, the differences in ARGs between two counties observed in this study could be attributed to soil type and manure type. Additionally, the transport and persistence of ARGs also might be influenced by crop planting and harvest.

Integron was composed of the *intI* gene and genes conferring resistance, connected by a variable region wherein one or more gene cassettes could be inserted [46]. Previous studies suggested that the *intI gene* be regarded as an indicator to evaluate the capacity of horizontal gene transfer through acquiring and disseminating ARGs [41]. Most integrons in the environment were of the class 1 type [46]. It was verified that the relative abundance of *intI1* was significantly higher than that of *intI2* by this study and the previous study in greenhouses [24]. Positive correlation of *intI1* with *cmlA*, *aadA2*, *sul1*, *sul2*, *tetC*, *tetG*, *and tetL* suggested horizontal gene transfer of these ARGs among microorganisms. This was also involved in multiple resistances, because the 3′-conserved segment of *intI1* gene cassette, where *sul1* is located, was detected in some other genes [46]. A large conjugative plasmid with *ANT(2″)Ia-aadA2* cassette located on class 1 integron [36] could also explain the positively correlation between *intI1* and *aadA2* in this study. In this study, the positive correlations among ARGs, such as *cmlA* with *aadA2*, and both of the above with *sul1*, *sul2*, and *tetG*, suggested that ARGs have some potential interactions, although they have encoded resistance to different types of antibiotics.

### 4.3. Potential Hosts of ARGs and intI Gene

The changes in bacterial community demonstrated essential roles in the evolution of ARGs by determining their potential hosts [47]. Network analysis indicated that 11 bacterial phyla might be the potential hosts of specific ARGs. For example, *Gemmatimonadetes* was the potential host of *cmlA*, *sul1*, and *sul2*; *Chloroflexi* was the potential host of *sul1* and *tetM.* It is interesting that although abundances of *Hydrogenedentes*, *Tenericutes*, and *Spirochaetes* were low, these were the potential hosts of t*etL* (the most abundant *tet* gene). However, previous study based on media culture found that almost all the identified tetracycline-resistant bacteria isolates carrying *tetL* belonged to *Firmicutes*, *Actinobacteria*, and *Proteobacteria* [38]. This might be because the methods used were different. Most of the soil bacteria are still unculturable, and only a few of the hosts were verified based on current media culture [9]. It was noted that *Gemmatimonadetes* was the potential host of *intI*, as well as, *cmlA*, *sul1*, and *sul2*, which suggested that horizontal gene transfer of *cmlA*, *sul1*, and *sul2* is carried by *Gemmatimonadetes.*

### 4.4. Contributions of Factors to Bacterial Community and ARGs

The dissemination and stability of ARGs in soil were affected not only by the selection pressure of antibiotics, but by the abiotic factors, including soil pH, organic matter, soil moisture, temperature, heavy metal, and micro/macro nutrient availability [48,49]. In this study, antibiotics, as well as soil pH, organic matter, and heavy metals were investigated. However, none of the 17 selected antibiotics were detected in any of the greenhouse soils except for doxycycline in soil from SG1. The persistence of antibiotics in the terrestrial environment was various, ranging from several hours to weeks or even months [23]. Thus, antibiotics were mostly degraded or biotransformed, because the sampling was conducted almost at the end of the cropping period, which was almost 9 months after manure application. In the meanwhile, diverse and abundant ARGs were discovered in all samples, even in the soil absent of antibiotics, implying that the presence of antibiotics is not be a precondition of the occurrence and persistence of ARGs/antibiotic-resistant bacteria in soil. Similarly, several studies found the common occurrence of ARGs in soil, even though no antibiotics were detected [50,51]. Further attention should be paid to the mechanisms behind the persistence of ARGs in soil with no antibiotic selection pressure, which may contribute to the further problem of the dissemination of resistance to antibiotics.

A significant proportion of bacterial community variation was explained by metals and pH value in farmland [52]. The heavy metal may cause a shift in the bacteria composition. Significant positive correlations were detected between microbial communities and heavy metals (Cr, Cu, Ni, and Zn) [53]. Additionally, pH and heavy metals have comprehensive effects on bacterial communities [54]. A relatively low pH increased the bioavailability of heavy metal and micro/macro nutrient availability, and exerted strong selective pressure on microorganisms [55,56]. This could explain why pH was one of the strongest environmental factors determining the overall bacterial communities in this study. Heavy metals promoted conjugative transfer of ARGs between microorganism strains [15], and enhanced the coselection of ARGs and metal resistance genes in soils with high heavy-metal contents [16]. However, the concentration of Cr, as well as Ni, Pb, Cu, As, were negatively correlated with most of the ARGs, including *cmlA*, *aadA*, *aadA2*, *sul1*, *sul2*, *tetA*, *tetL*, and *tetM* in this study, suggesting that the selective pressure of these heavy metals on ARGs at low concentrations should be taken seriously, rather than be ignored. Ni, as well as Cr and Pb, were positively correlated with *tetB(P)*. At the meantime, Ni, Cr, and Pb were positively correlated with *Proteobacteria*, while *Proteobacteria* was positively correlated with *tetB(P)*. Thus, factors, such as Ni, Cr, and Pb affected the distribution of ARGs not only by themselves, but by affecting the structures of bacterial communities.

## 5. Conclusions

In summary, this study provided a useful insight of the profiles of bacterial communities and ARGs in vegetable greenhouse soils with different manure-application years and locations. The dominant bacterial phyla were *Proteobacteria*, *Firmicutes*, *Acidobacteria*, *Chloroflexi*, and *Gemmatimonadetes*, and the most abundant ARGs were *aadA*, *tetL*, *sul1*, and *sul2* in greenhouse vegetable soil. The positive correlation between the *intI1* gene with *cmlA*, *aadA2*, *sul1*, *sul2*, *tetC*, *tetG*, *tetL*, as well as that between *intI2* with *strA*, *strB*, highlights the horizontal transfer potential of these ARGs. The changes in bacterial community play essential roles in the evolution of ARGs by determining their potential hosts. Ni and pH were the most potent factors determining the bacterial communities, and Cr was the top factor affecting ARGs. Further study on the characterization of antibiotic-resistant bacteria, as much as possible, and a metagenomic survey should be carried out to elucidate the mechanisms behind the occurrence and persistence of ARGs in soil in special microclimate environments, such as vegetable greenhouses.

## Figures and Tables

**Figure 1 ijerph-19-07742-f001:**
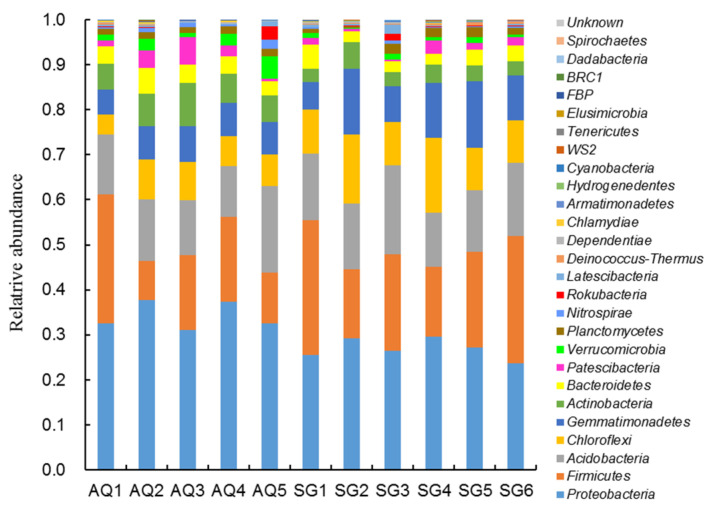
The profile of bacterial communities in different greenhouse soils (phylum level).

**Figure 2 ijerph-19-07742-f002:**
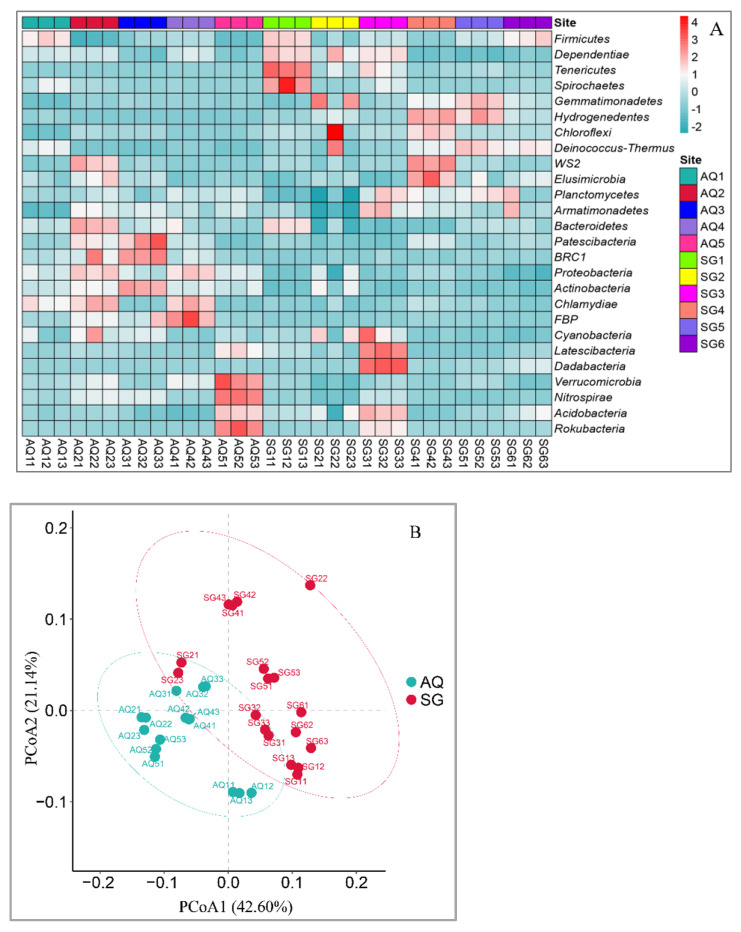
Heatmap (**A**) and PCoA (**B**) based on the Bray–Curtis index of bacterial communities in different greenhouse soils.

**Figure 3 ijerph-19-07742-f003:**
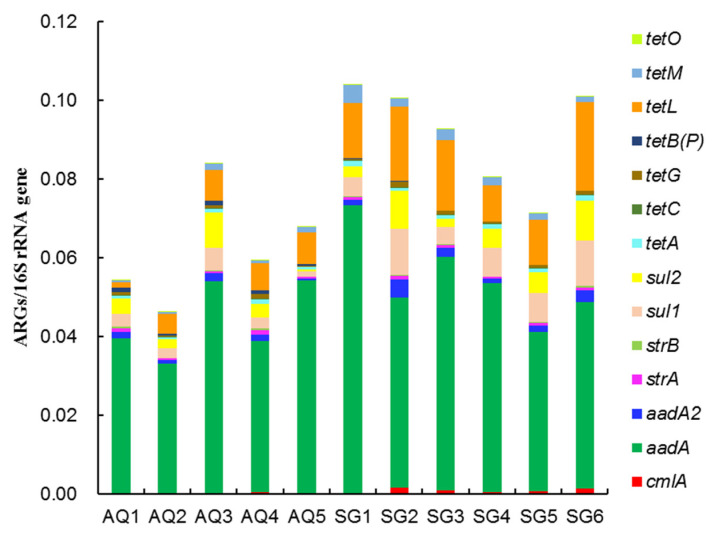
Relative abundances of ARGs in different greenhouse soils.

**Figure 4 ijerph-19-07742-f004:**
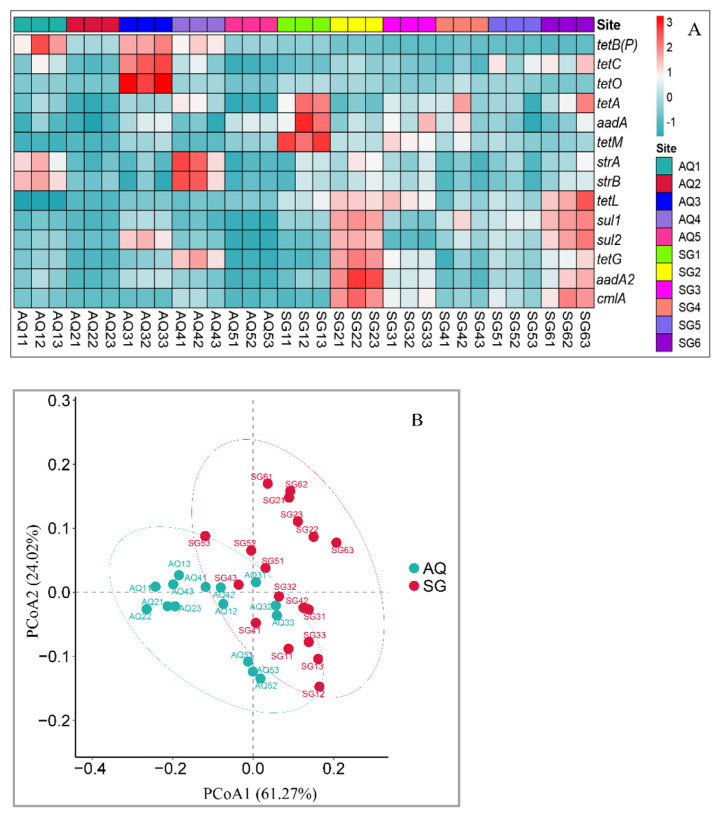
Heatmap (**A**) and PCoA (**B**) based on the Bray–Curtis index of ARGs in different greenhouse soils.

**Figure 5 ijerph-19-07742-f005:**
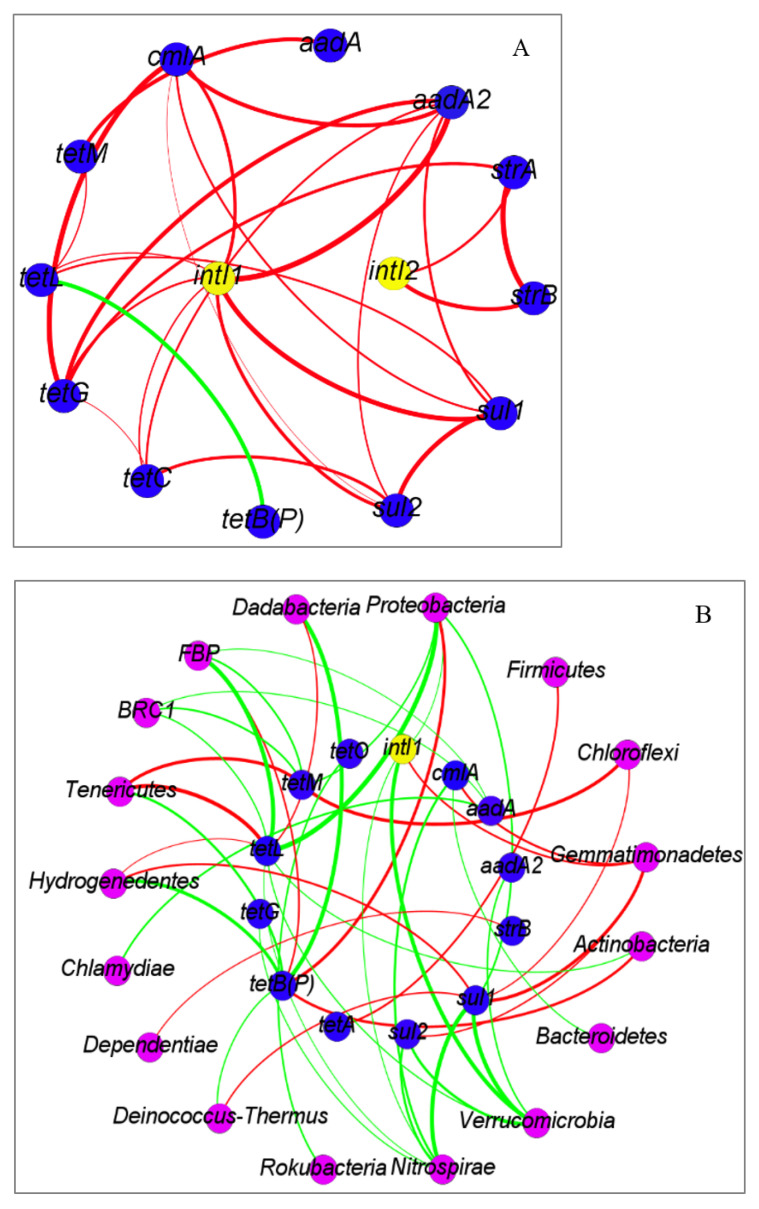
Network analysis of correlation ((**A**)—the relationship of ARGs and *intI* genes; (**B**)—the relationships of ARGs and *intI* genes with bacteria). Edges are weighted according to the correlation coefficient. Red edges mean positive correlation, green edges mean negative correlation. Only significant correlation is shown (*p* < 0.05).

**Figure 6 ijerph-19-07742-f006:**
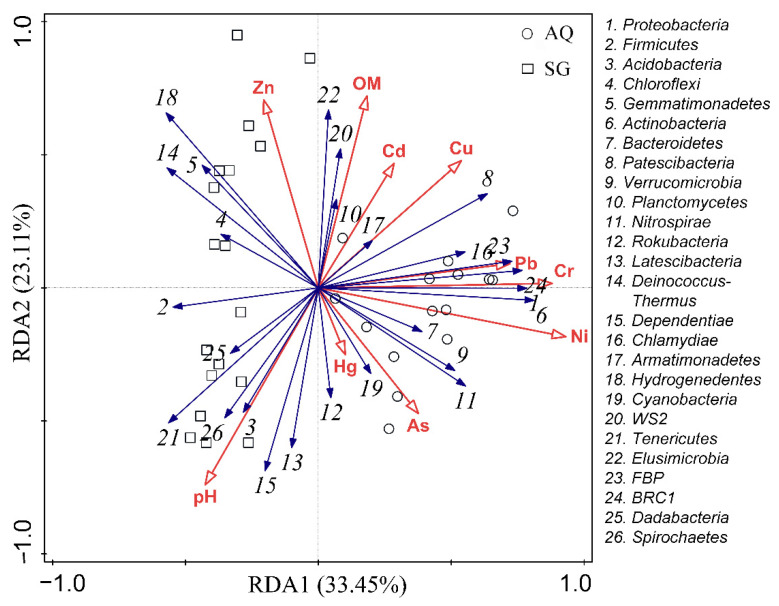
RDA of bacterial communities in different greenhouse soils.

**Figure 7 ijerph-19-07742-f007:**
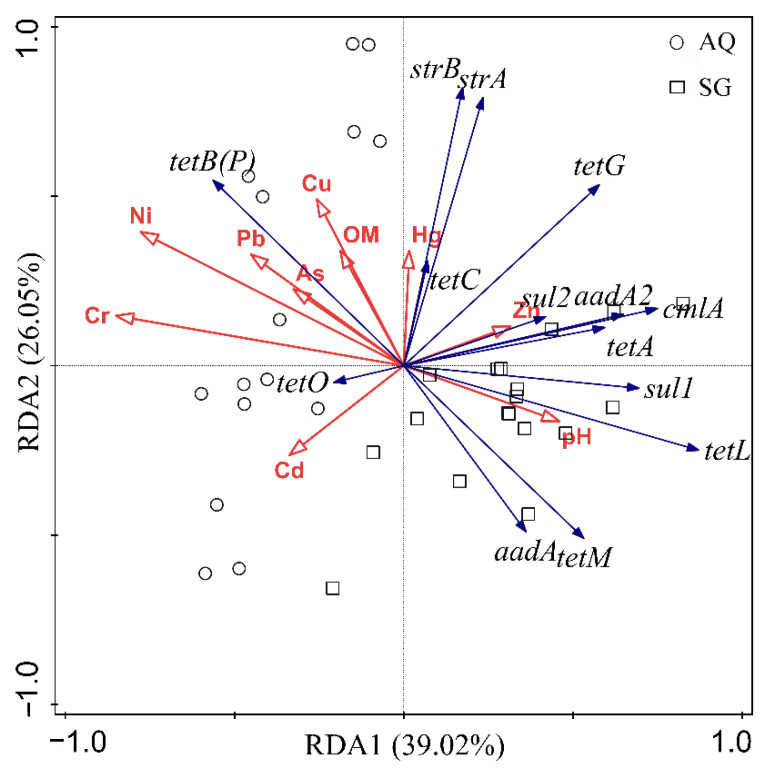
Redundancy analysis (RDA) of ARGs in different greenhouse soils.

**Table 1 ijerph-19-07742-t001:** Contributions of physicochemical properties and heavy metal concentrations on bacterial communities and ARGs based on RDA.

Factors	ARGs		Bacterial Community	
	Contribution %	*p*	Contribution %	*p*
Cr	29.2	0.002	2.1	0.46
Cu	12.2	0.006	7.9	0.004
Cd	12.4	0.006	13.5	0.002
OM	11.7	0.002	12.1	0.002
pH	10.3	0.004	17.7	0.002
Ni	8.0	0.01	30.6	0.002
Zn	7.6	0.016	6.5	0.008
As	3.9	0.092	4.9	0.032
Pb	2.4	0.238	0.9	0.896
Hg	2.3	0.226	3.9	0.092

## Data Availability

Not applicable.

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
