# Peer review of "Profile of Bacterial Community and Antibiotic Resistance Genes in Typical Vegetable Greenhouse Soil"

_ijerph, 2022, doi:10.3390/ijerph19137742_

Round 1
Reviewer 1 Report
- Author should be added the information and background of sample size location in the methodology part.
- Author should be added the reference source of DNA extraction.
- Author should be added the conclusion table for Contributions of factors to the ARGs and bacterial community with the reference source.
Author Response
Dear reviewer,
Thank you for your comments on our manuscript entitled “Profile of antibiotic resistance genes and bacterial community in typical vegetable greenhouse soil” (Manuscript ID: ijerph-1756985). Those detailed and constructive comments are very helpful for improving our paper.

Reviewer 2 Report
Comments and suggestions for improvement can be found in the pdf file attached
- further improvement of the English writing style and grammar should be made before resubmission (introduction & discussion sections). Have a look at the corrections/suggestions made to the abstract
- new and significant findings in the results should be elaborated and discussed in details in the discussion section

Author Response

(The authors gave the same response as above.)
